# Real-time classification of longitudinal conveyor belt cracks with deep-learning approach

**Uttam Kumar Dwivedi** [ID]*, **Ashutosh Kumar** [ID], **Yoshihide Sekimoto**

Department of Civil Engineering, The University of Tokyo, Tokyo, Japan

* uttamdwivedi04@gmail.com

**Data Availability Statement:** Data have been uploaded to the following DOI: (https://osf.io/vm3wt/?view_only=95c5e9423b5a413dbff6931a92ce2195).

## Abstract

Long tunnels are a necessary means of connectivity due to topological conditions across the world. In recent years, various technologies have been developed to support construction of tunnels and reduce the burden on construction workers. In continuation, mountain tunnel construction sites especially pose a major problem for continuous long conveyor belts to remove crushed rocks and rubbles out of tunnels during the process of mucking. Consequently, this process damages conveyor belts quite frequently, and a visual inspection is needed to analyze the damages. Towards this, the paper proposes a model to configure the damage and its size on conveyor belt in real-time. Further, the model also localizes the damage with respect to the length of conveyor belt by detecting the number markings at every 10 meters of the belt. The effectiveness of the proposed framework confirms superior real-time performance with optimized model detecting cracks and number markings with mAP of 0.850 and 0.99 respectively, while capturing 15 frames per second on edge device. The current study marks and validates the versatility of deep learning solutions for mountain tunnel construction sites.

## 1. Introduction

While Japan holds a long running legacy of construction engineering worldwide, it is worth to mention that tunnels have been an important part of their road and railway network due to complex geology and dense urban areas. As of recent data, Japan has approximately 5180 kilometers of road tunnels and 3,813 kilometers of railway tunnels [1, 2]. With the advent of new tunnel technology [3, 4], there is a rekindled interest for sustainable development by enabling short construction period, cost reduction, environmental preservation, and quality improvement. Among these, the New Austrian Tunneling Method (NATM) [5, 6] is frequently used as the basis of modern tunneling technologies. In this method, mucking is performed to carry out crushed rocks (muck) on long conveyor belts after explosives are ignited and detonated to break the rocks at the tunnel face [7] as shown in Fig 1. However sharp rocks of different sizes generally damage the conveyor belts, eventually penetrating it through after dropping from trail trolly, which generally prevent the continuous belt conveyor from working properly,

**Funding:** This work was supported by the financial support from Tokyo Kizai Kogyo co. ltd. (http://www.tokyokizai.com/). The funders had no role in study design, data collection and analysis, decision to publish, or preparation of the manuscript.

**Competing interests:** The authors have declared that no competing interests exist.

causing accidents or belt rupture [8]. As a result, it's important to frequently inspect the belt surface to look for damage and catch it early.

Further, the above griming problems demands new inexpensive and sustainable developments, where the safety engineers are responsible for inspection of conveyor belts in mountain tunnel construction continue to depend on visual examination by halting the mucking process. The manual visualization like this creates two major problems. First, the inspectors are under a great deal of strain due to long belts and hard work hours involved in the inspection, which causes problems such as overlooking damage due to fatigue and reducing the frequency of inspections. Second, stopping the conveyor belt during inspections reduces the work efficiency of mucking process, which leads to delays in the entire work process. Overall, this imposes open research on "How can edge AI-based deep learning framework be used to detect and track damages on long conveyor belts in real-time without halting the mucking process and ensure safety and productivity at mountain tunnel construction sites? These problems call for the development of a system that can automatically detect damage in real-time without placing a burden on any engineers or construction workers for performing the inspection and notify site engineers about the exact location and type of damage without needing the mucking process to stop.

In pursuit of the above scenario, the current work proposes an edge AI based deep learning framework consisting of three parts. First, detect and track the damages along with its type such as small, medium, large, or through in long conveyor belt. Second, identify the location of damage by detection of three number markings representing length marking at every 10 meters of the conveyor belt. Third, provide real-time alerts to safety engineer via offline web server within edge device. Proposed platform uses on-device processing to ensure real-time detection and localization of damages on conveyor belts moving at a speed of 120 m/min and 180 m/min.

The effectiveness of the proposed method has been tested on 3 different mountain tunnel construction site data using an Nvidia Jetson NX edge device [9] with monocular USB camera. The result showed that the average overall mean average precision (mAP) of proposed damage detection and localization system are 0.85 and 0.99 respectively, therefore has a potential to enhance productivity and safety at mountain tunnel construction sites. Overall, this work uses a deep learning-based solution and image processing on offline edge device to establish real-time alerting platform for damages on long conveyor belt.

## 2. Literature review

Automatized inspection of tunnel constructions constitutes one of the grim yet interesting field for researchers working in the construction field [10–12]. Over the years, many devices have been developed to address these issues. For example, infrared thermal imaging technology [13, 14] has been devised to overcome dusky and dark underground environment of coal mines, where spectrum features are extracted from 2D spectrum signals obtained by Fast Fourier transform of the conveyor belt images. For example, Qiao et al. [15] proposed a binocular visual detection method using visible light to extract scene and infrared light to extract edge features. The length, width, and area of longitudinal tears are obtained from the projection vectors of the acquired images on the X and Y axes. Similarly, X-ray based nondestructive techniques (NDT) [16, 17] were suggested, where high penetration characteristics of X-rays are used to identify large damages. Although this method can detect cracks but require specific image processing with high-end expensive setup and specially focus on large damages on conveyor belts used in coal mines.

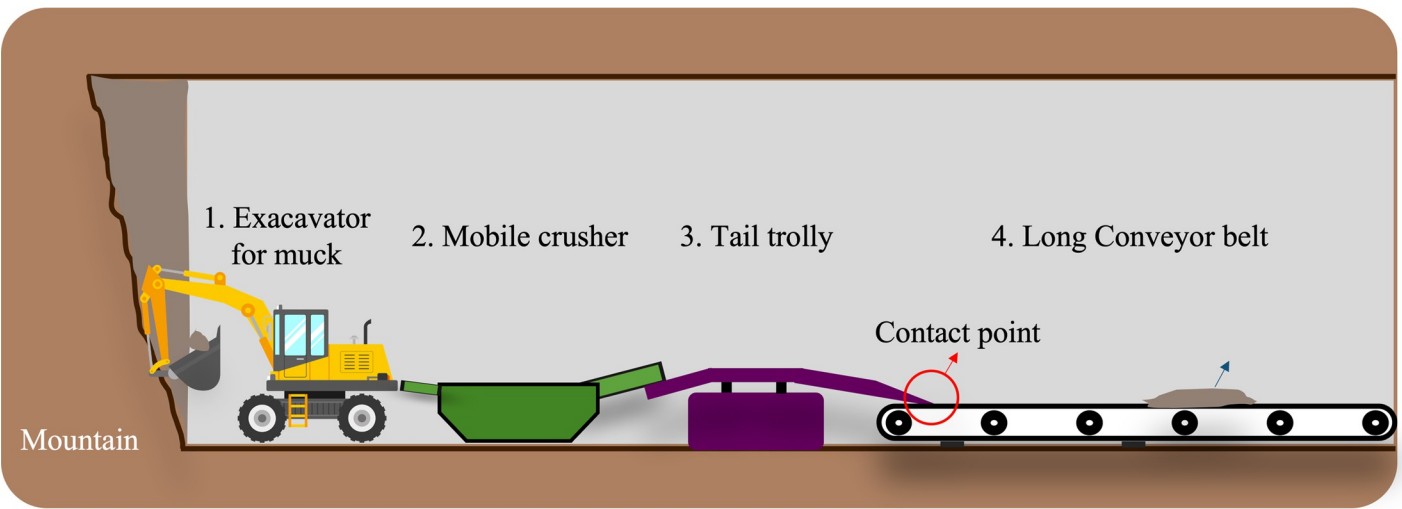

**Fig 1. Muck transfer by continuous long conveyor belt method in mountain tunnels.**

AI based methods especially deep learning algorithms have been widely used in image classification, detection and segmentation have been employed, where high-speed CMOS cameras are used in combination with high performance computing devices to extract features from images [18, 19]. Guo et al. [20] proposed YOLOv5 based object detection method to detect and locate conveyor belt damage region in real-time. Agata et al. [21] applied easy to setup MATLAB's deep learning solution with two-layer neural network. These methods provide good balance of network architecture in depth and image resolution while providing adequate detection speed and mean average precision (mAP). Proposed model locates the damage of different sizes as well as identify the numbers marked at the side of conveyor belts. These markers indicate the distance of the conveyor belt, which makes the repair easy.

## 3. Research methodology

The methodology to frame proposed study is focused on two main areas. First, preparation of proper experimental setup for data gathering to include various scenarios and second, to use computer vision techniques to identify and localize cracks on long conveyor belts.

### 3.1 Experimental setup and data collection

**3.1.1 Data collection.** In this research, we develop the Conveyor Belt Crack Detection (CBCD) dataset consisting of 9,362 images. The images were collected from mountain tunnel construction sites, experiment setup stations and using web crawling techniques [22] in collaboration with Tokyo Kizai Kogyo co. ltd. Out of 9,362 images, 1562 images have conveyor belt cracks with handwritten number markings as shown in Fig 2, while the rest 7,800 images have no cracks. On all 7,800 images, we superpose 70,000 handwritten digits from the MNIST [23] dataset, as shown in Fig 2.

Essentially, the MNIST database is a large opensource database of handwritten digits that is commonly used for training various image processing systems. It is done to increase the number of samples for digit recognition for the localization of crack since the number of images with handwritten digits are not enough. The CBCD dataset contains 11 classes which include the digits 0 to 9 (Fig 2(D)) and crack class (Fig 2(B) and (2C)). We randomly split the CBCD dataset and use 8,188 images for training set and 1,174 images for validation.

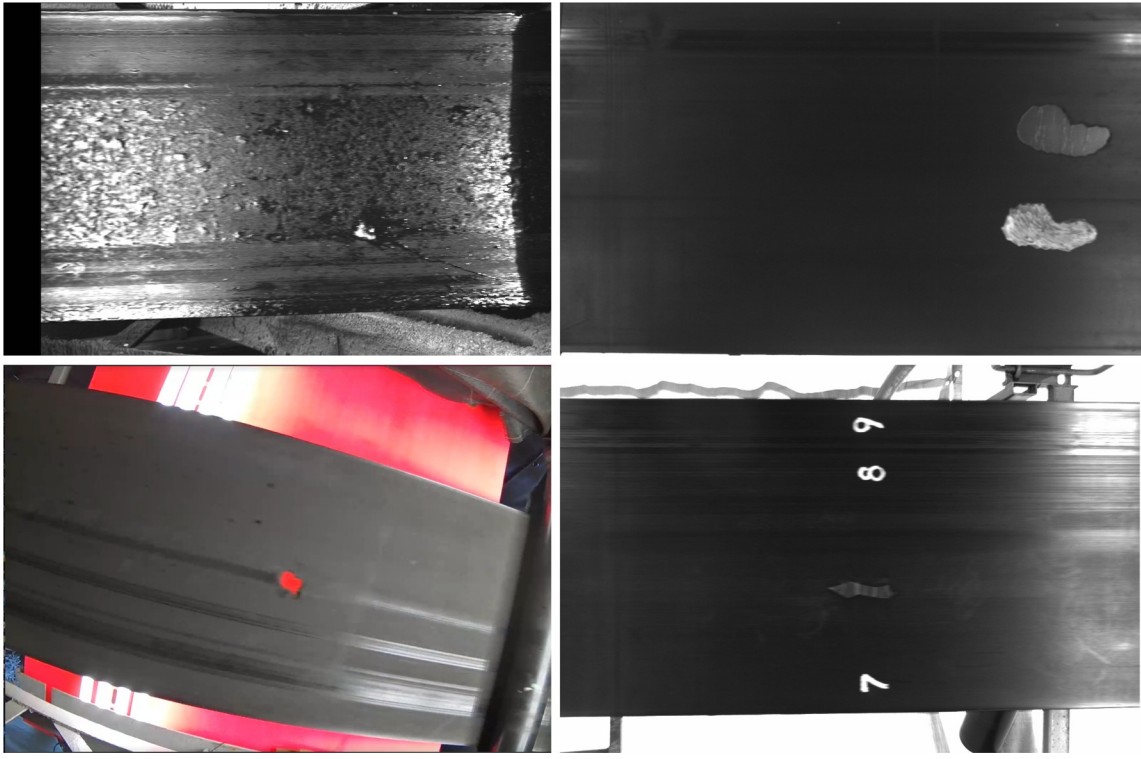

**Fig 2. Training dataset for Conveyor belt crack detection (CBCD).** (A) shows reflecting surface without any cracks, (B) shows non-through cracks while (C) shows through cracks with orange lights reflecting through it.

**3.1.2 Experimental setup.** Fig 3 illustrates a schematic diagram of the setup device. The camera was installed so that it faces upward from the bottom of the conveyor belt, and distance of 1.5 m was set up between the tip of the camera and the belt. This distance is intended for actual installation at a construction site. The conveyor belt to be filmed was washed by a stream of water so that no contamination from the sleds transported would remain on the surface of the belt. The camera was installed in a dark room covered with a protective sheet, and lighting was installed next to the camera to ensure good-lighting condition for camera. Second LED light was installed of orange color to differentiate through damage type as it will have orange lights coming on the other side for the camera to capture. Conveyor belts used in mountain tunnel construction sites have a width of 0.6 m and a three-layer structure consisting of a rubber layer, a polyester layer, and a rubber layer, and each is approximately 10 mm thick.

## 3.2 Applied computer vision techniques

This section devotes explanation of deep learning models, image processing techniques used in proposed paper.

**3.2.1 Deep learning-based detector model.** The essence of long conveyor belt damage detection is target detection. Proposed paper uses single stage target detection algorithm YOLOv4 [24] as deep learning model for object detection network to detect conveyor belt cracks and number markings written on the belt. Since the original YOLOv4 model is used to detect objects on the COCO dataset [25] with 80 classes, the network architecture was modified to incorporate proposed 11 classes. We train YOLOv4 using the 8,188 images from the CBCD dataset for 40,000 iterations with a batch size of 64 using the initial pre-trained weights from

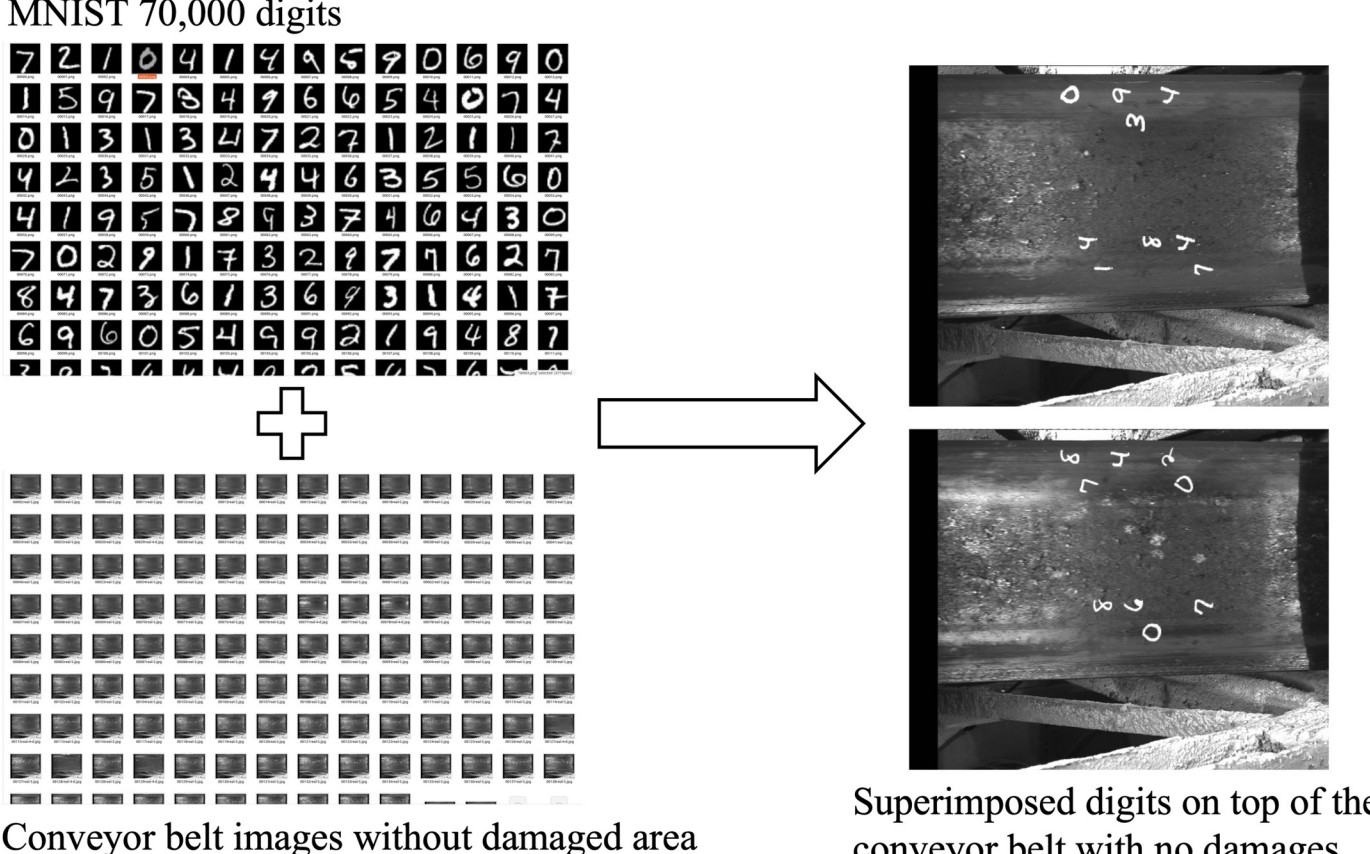

**Fig 3. Handwritten digits from MNIST dataset are superimposed on top of conveyor belt images.** Conveyor belt areas with no damages are chosen for superimposition to increase the number of samples for digit recognition for the localization of the crack on the conveyor belt.

ImageNet dataset [26] for the first 137 convolutional layers. For training, two NVIDIA GeForce RTX 3090 with 24 GB GPU memory were used. The hyperparameters for training is shown in Table 1. The training loss [27] and mean average precision (mAP) [28] is shown in Fig 4.

**3.2.2 Optimization of the neural network.** Further, optimization of the YOLOv4 network is necessary for the real-time processing on lightweight edge devices. Neural networks generally use FP32 floating point precision [29] for storing parameters such as weights and biases. Using higher precision increases computational complexity and increases the size of the model. Experimentally, it has been seen that a neural network model with half-precision FP16 for the parameters has similar performance as that with single-precision FP32. Therefore, the precision can be reduced to FP16 without compromising much on the performance. This

**Table 1. Hyperparameters of the YOLOv4 network for training on the CBCD dataset.**

| Hyperparameter | Value |
|---|---|
| Input size | 608 |
| Learning rate | 0.001 |
| Batch size | 64 |
| Sub-division | 16 |
| Optimizer | SGD with momentum |

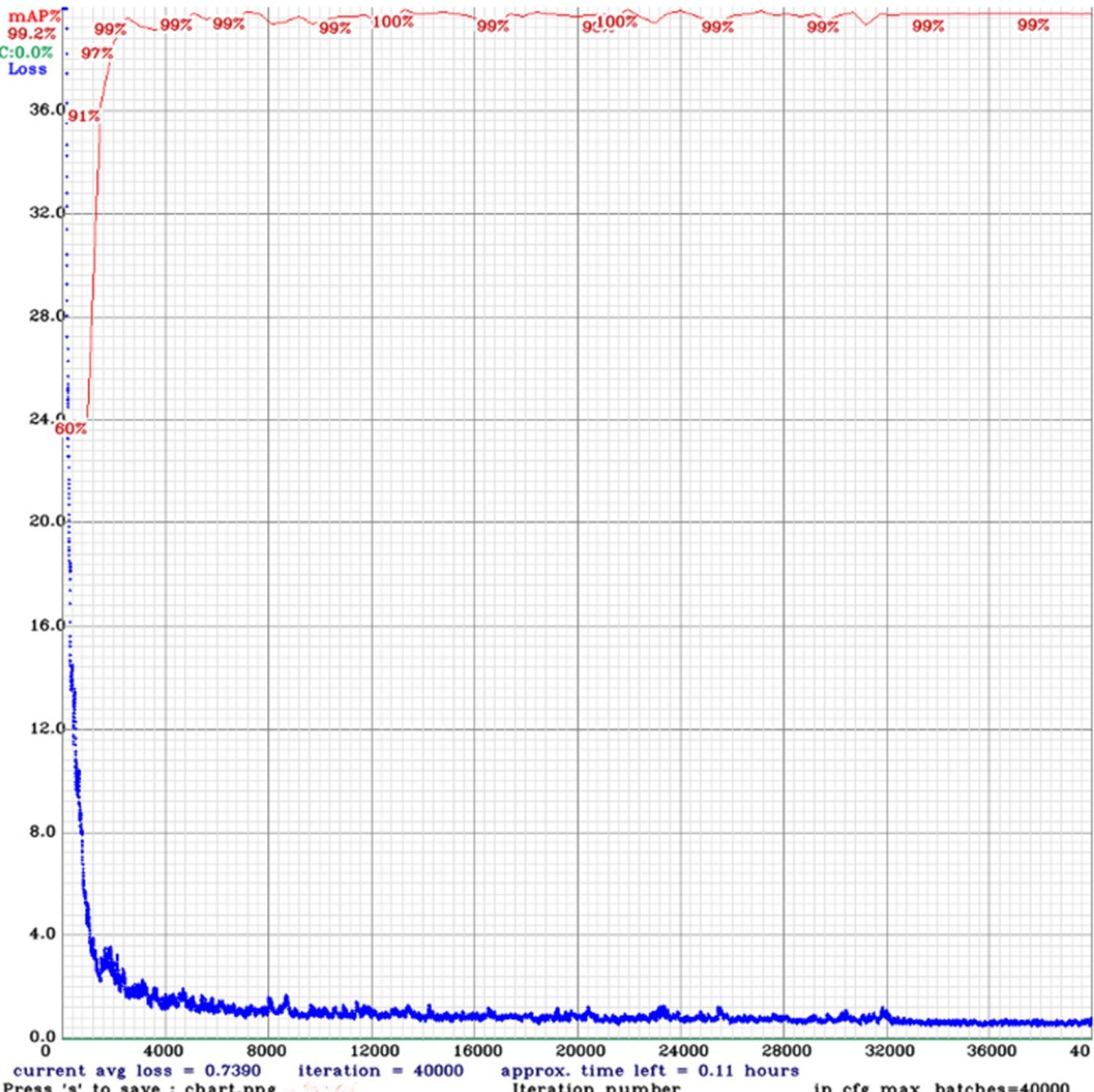

**Fig 4. Training loss (Blue) and validation mAP (red) variation for trained deep learning model.** X axis represents the number of iterations.

could be attributed to the fact that neural networks are quite resilient to the noises. A reduction in the precision value from FP32 to FP16 is seen as the introduction of the noise. Further, half-precision models are very light compared to the single-precision model and has significant increase in the inference speed [30]. We carry out optimization in the TensorRT framework [31] by reducing floating point precision to FP16 and fusing layers that perform routine operations, as shown in Fig 5.

**3.2.3 Crack detection and localization.** Next, we deploy the optimized TensorRT model on the edge device Jetson NX for the detection of cracks on conveyor belt. The target conveyor belt of the experiment area is shown in Fig 6 with the specifications of the conveyor belt. Generally, it is difficult to accurately detect cracks since the environment around the conveyor belt maybe quite different (e.g., uneven light, dirty surface, etc.). To accurately detect the cracks, we

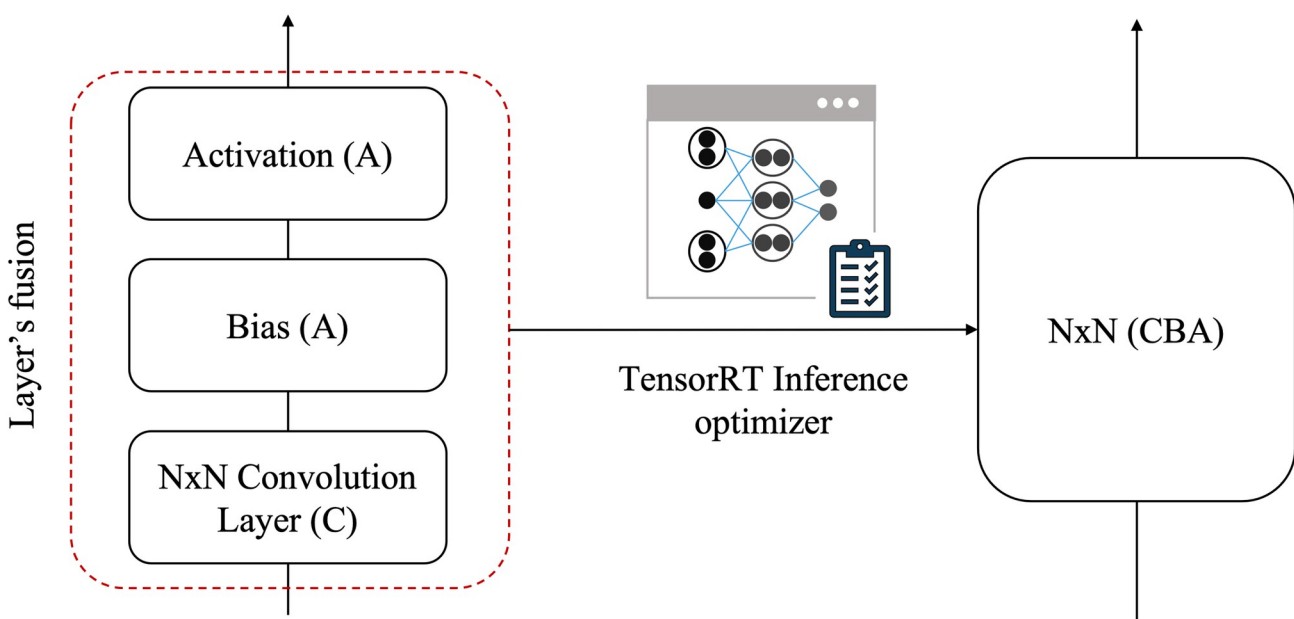

**Fig 5. Fig shows the horizontal and vertical fusion of layers in the TensorRT framework.** It describes that NxN convolutional layer (C), Bias (B) and Activation layer (A) are combined to form a single block NxN (CBA).

put an orange LED light strip behind the conveyor belt and put the camera with white light source focusing on the belt, as shown in Fig 6. The main advantage of using such an approach is that when there is a crack in the belt, the orange color light passes directly through it, which can be easily detected by the crack detection model.

The conveyor belt also has three digits written on top as shown in Fig 2(D) at a fixed interval of 10 meters for number marking in a unique combination that a particular number only

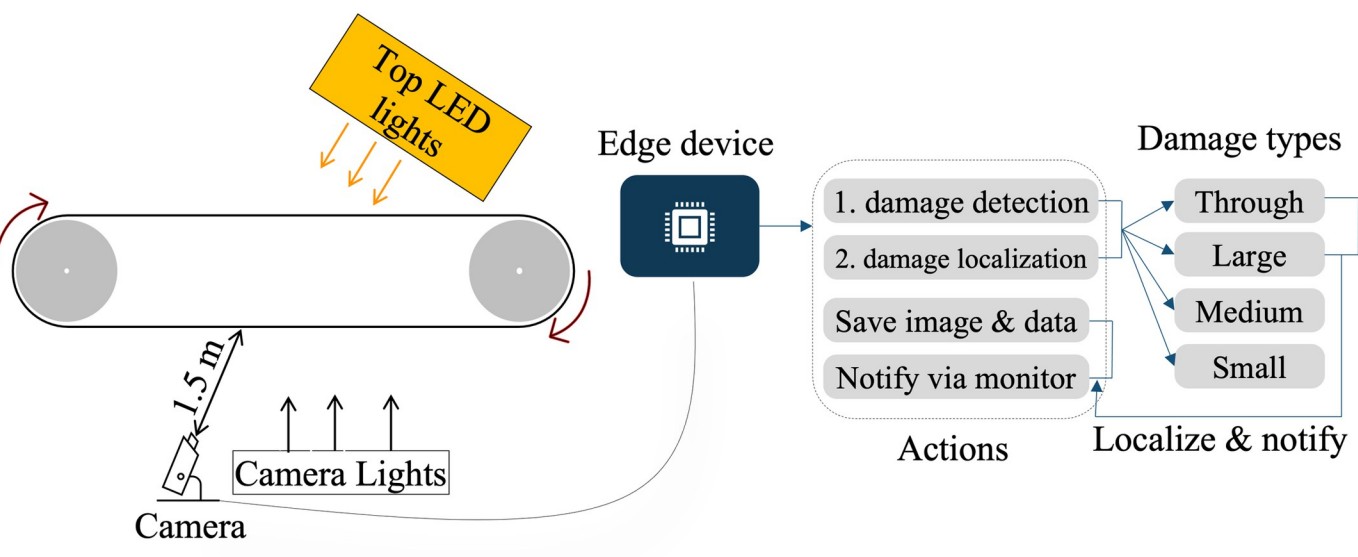

**Fig 6. Simplified diagram of experiment setup and expected output.** Edge device's deep learning and image processing program provides output. Top LED orange lights provide contrast with respect to bottom white lights.

appears once in the entire belt. To localize the crack i.e., to find the location of the crack on the conveyor belt, we detect the numbers as they appear on the belt. Once the numbers are detected, we store them as the first location point. Assuming a crack is detected after the appearance of first numbers, we record the number markings appearing after the detection of crack. In this way, the location of the crack on the conveyor belt can be located.

**3.2.4 Crack size estimation.** Finally, we also estimate the size of the crack in metric units using the monocular camera. Our approach involves the distance estimation technique proposed by Karney et al. [32]. Essentially, their approach dwells on estimating the distance of an object if the true dimension of the object is known, and provided that focal length, camera sensor dimension and image resolution is fixed. In our approach, however, instead of estimating the distance of the crack from the camera, we fix the distance of the camera from the conveyor belt as shown in Fig 6. The only parameter remains to be determined, in this case, is the crack dimension, which can be evaluated using Eq 1.

$$\mathcal{H}_{crack}(\text{in mm}) \ = \ \frac{d \ \times \ h_{crack,px} \ \times \ \mu_h \ \times \ 1000}{f \ \times \ I_h} \tag{1}$$

[Where $\mathcal{H}_{crack}$ = Crack size in metric units (mm); d = Fixed distance of the conveyor belt surface from the camera; $h_{crack,px}$ = Height of the crack in pixels obtained from the bounding box; $\mu_h$ = Height of the camera sensor; f = Focal length of the camera; $I_h$ = Height of the image resolution]

# 4. Results

Three different mountain tunnel construction sites data were selected as a test bed to identify cracks on long conveyor belts. We trained the YOLOv4 model using the CBCD training set containing 8,188 images. The mAP of the trained model after 40,000 iterations and the individual AP per class on the validation set containing 1,174 images is presented in Table 2.

Next, we optimize the model using TensorRT framework and re-evaluate the AP per class of the optimized model. The inference speed and comparison of AP for each class is shown in Fig 7. From Fig 7, we observe a significant increase in the speed of the optimized model on the edge device Jetson NX thus improving frames per second (FPS) from 5 FPS to 15 FPS, while keeping mAP very close to original model as shown in Fig 8.

**Table 2. Table showing the average precision (AP) of crack and various classes of digits for number markings.**

| Class | AP |
|---|---|
| crack | 0.85 |
| 0 | 0.99 |
| 1 | 0.99 |
| 2 | 0.99 |
| 3 | 0.99 |
| 4 | 0.99 |
| 5 | 0.89 |
| 6 | 0.99 |
| 7 | 0.99 |
| 8 | 0.99 |
| 9 | 0.99 |
| **mAP** | 0.99 |

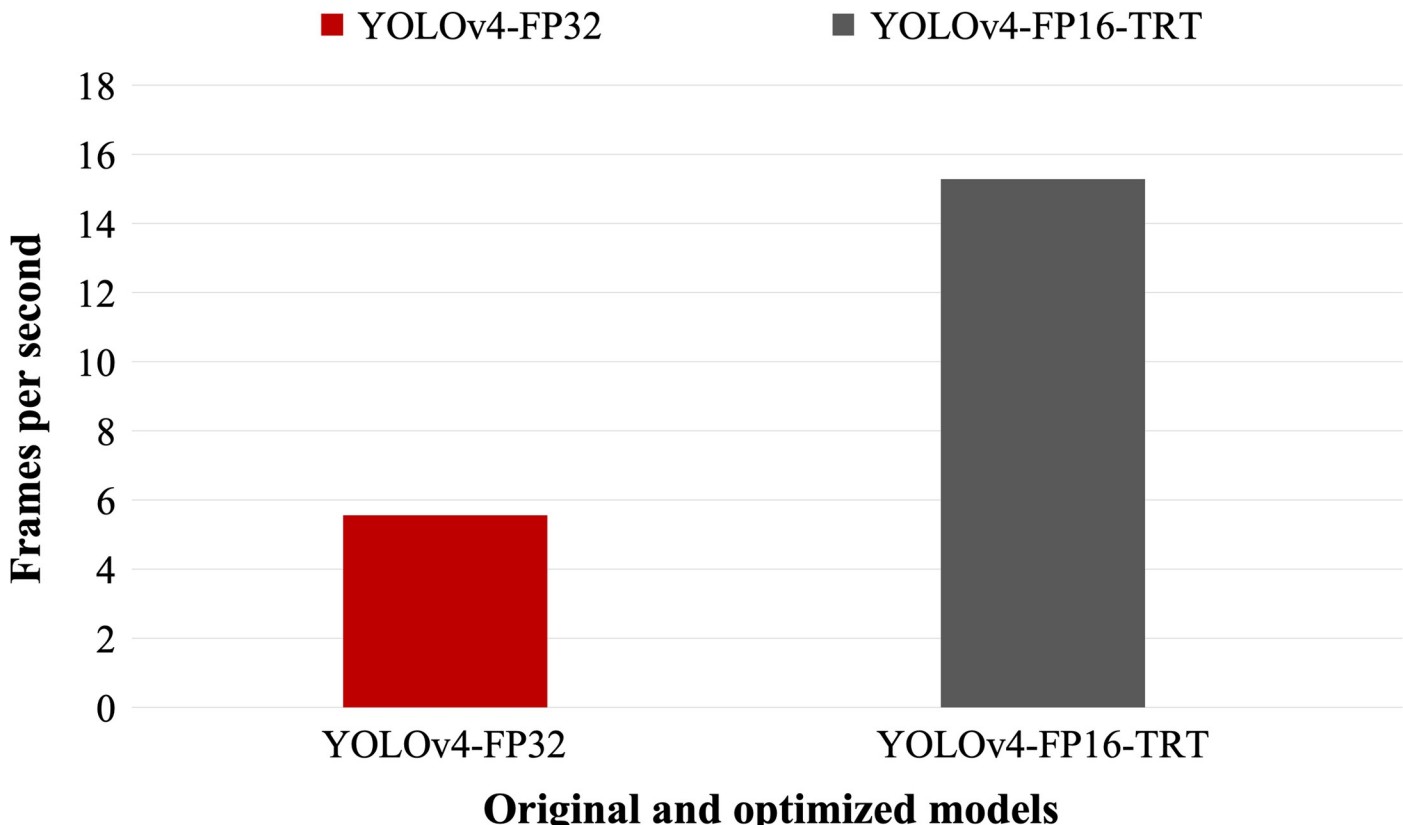

**Fig 7. The comparison of inference speed of the original YOLOv4 model and its optimized version.** Comparison of YOLOV4(YOLOv4-FP32) with 608x608 input resolution and its optimized version in TensorRT (YOLOv4-FP16-TRT). The frames per second (fps) is calculated by averaging inference fps for 5,000 iterations.

### 4.1 Crack and number detection

500 test image samples of conveyor belt were collected across the mucking process of mountain tunnel construction site. Output sample of crack and number marking detection are shown in Fig 9, while the result is shown in Fig 8.

### 4.2 Crack detection results based on size

In Table 3, we show the accuracy of crack detection by its size. The results presented in Table 3 are based on crack detection results carried out at the actual site using Jetson Xavier NX device. We collect the samples from the image frames of the moving conveyor belt. Thus, the samples of damages show consider the same cracks at different locations and angles as the belt moves. We notice that a very small false positive for no damages, while the accuracy of crack detection reduces as the size of the crack reduces. We achieve the highest accuracy of 89.23% for large damages and the lowest accuracy of 64.13 for smaller damages.

## 5. Discussions

In general, conveyor belts are used in various fields, ranging from construction work to mining operations [33]. Their main use is felt for transporting rocks and gravels to several kilometers. The transported rocks and gravels are rather heterogeneous and include large as well as sharp rock pieces, which might cause longitudinal tears of various sizes. Due to the working

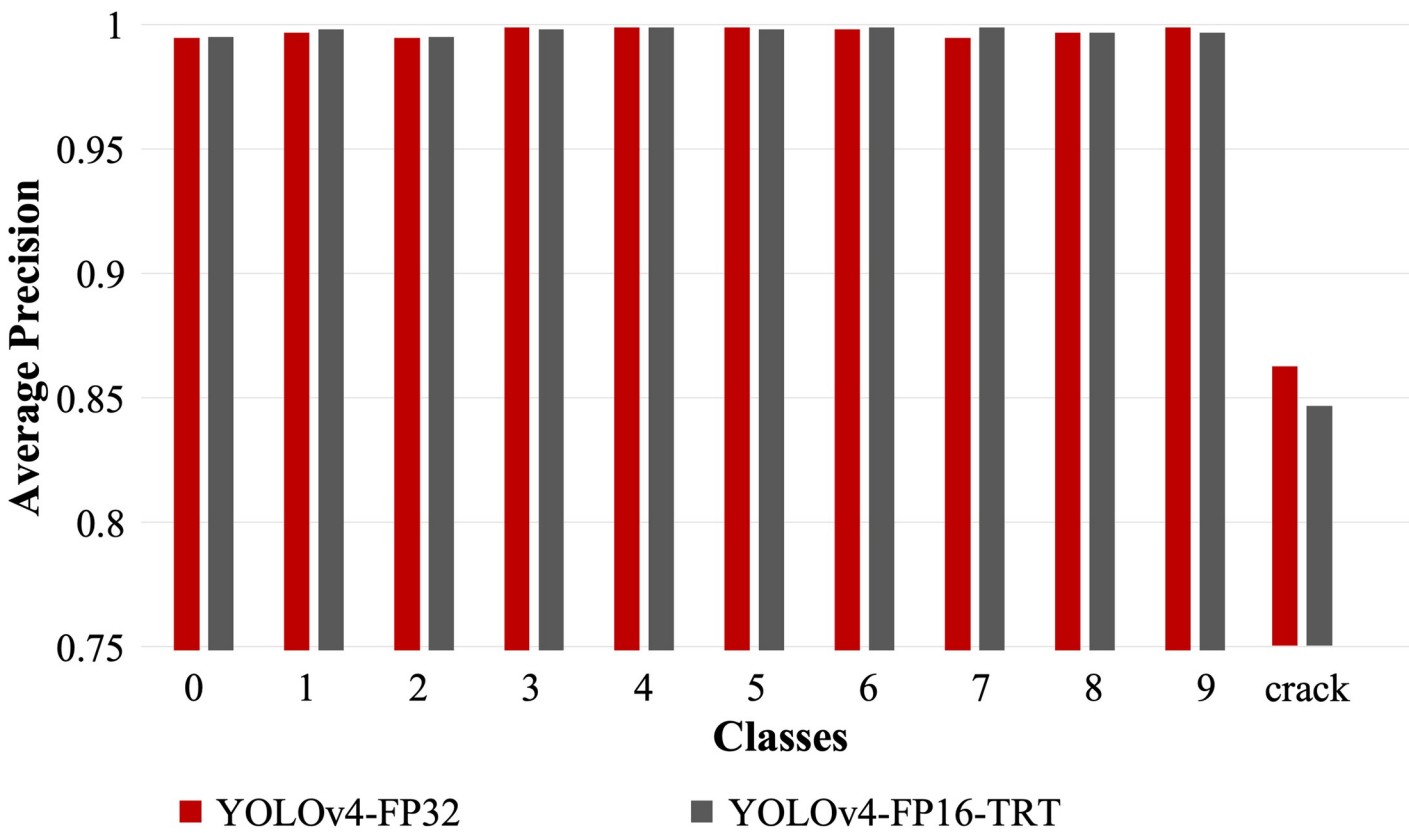

**Fig 8. The comparison of Average Precision (AP) of the original and optimized model.** Average precision is compared between original YOLOv4 model and the optimized model in TensorRT framework.

conditions inside tunnels as well as mines, finding out exact locations of damage by manual visual inspection in these long conveyor belts is significantly lengthy and expensive process. There are existing devices to solve this problem that use infrared laser and x-ray radiation [13, 14], ultrasonic and electro-magnetic energy probe [34] and image recognition software. However, these devices are very expensive and good in identifying large damages only. In contrary, we propose a simple, inexpensive, and sustainable study on detecting large as well as small damages by identifying number markings at every 10 meters of conveyor belt to alert the workers about the exact location that requires repair.

From the object detection results, we find that both crack and numbers can be detected with good accuracy. The numbers can be detected with almost perfect accuracy. This is due to advances in convolutional neural networks that can effectively learn all the features for different digits. From other research studies, we find that even shallow neural networks such as MobileNet [35] can achieve accuracy greater than 98% for digit classification. Crack detection results is lower compared to digit detection for number markings. This could be because cracks have more complicated features, which is harder to learn by the network. However, our crack detection mAP is similar to that presented in other studies. For example, study conducted by Guo et al. [20] achieved mAP of 82.5% using YOLOv5-m [36] for belt wear detection for large size damages.

To check the novelty of our proposed model, we reviewed the literature with some existing notable reports and compared with accuracy of our model in Fig 4. While there have been decent studies on welding work and outdoor crane work identifications, the proposed model

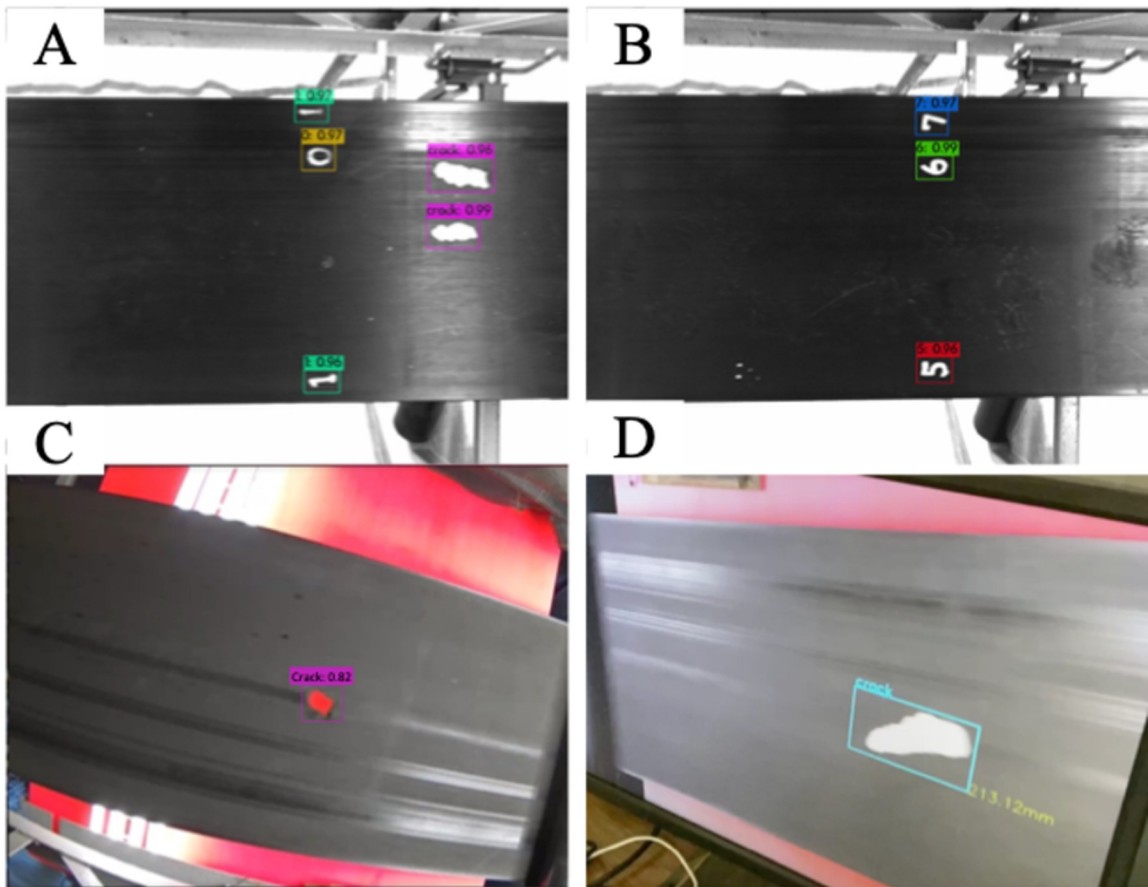

**Fig 9. Example of successful detection of digits and crack detection.** Digit detections are shown for number marking in (A), (B) and crack detection in (A), (C), (D) with corresponding size estimation (D) of the damage.

creates a niche in this direction with benchmarking accuracy of more than 85%. For belt tear detection, a vision-based method developed by Guo et al. [20] detects large size damages using YOLOv5-m [36] with a mAP of 82.5%. Similarly, Agata et al. [21] proposed an artificial intelligence-based approach for the classification of conveyor belt damage using two-layer neural network and reaches an accuracy of 80% and another method based on Haar-Ada Boost and Cascade algorithm was proposed by Wang et al. [37]. Where, longitudinal tear of a conveyor belt under uneven light were detected with an accuracy of 97%. However, these methods can detect only large type of damage, while proposed method can detect various types of damage, and the overall detection accuracy is improved as shown in Table 3.

**Table 3. Table shows the accuracy of the detection for various crack sizes.**

|  | No. of samples | Detected | Not Detected | Accuracy (%) |
|---|---|---|---|---|
| Large damage | 103 | 92 | 11 | 89.320388 |
| Medium damage | 120 | 91 | 29 | 75.833333 |
| Small damage | 92 | 59 | 33 | 64.130435 |
| Through damage | 34 | 28 | 6 | 82.352941 |
| No damage | 500 | 4 | 496 | 99.2 |

We find that the model YOLOv4-FP16-TRT after optimization of the neural network is 178.18% faster than the original YOLOv4-FP32 network. The reason for this is mainly due to the optimization techniques; particularly, precision reduction that greatly reduces the computational complexity, thereby increasing the inference speed. Despite the optimization of the neural network, in particular precision reduction, we do not notice significant decrease in the accuracy. This is mainly because reduction in accuracy causes the parameters (weights and biases) value to get truncated by the maximum supported by the FP16 precision. However, such truncations and reduction in precision is simply seen as the introduction of noise to which the neural network is quite resilient [38]. For the number detection, we notice the mAP of the digit detection is similar as the original YOLOv4-FP32 network. As mentioned before, this is because digit recognition is an easy problem and even simpler neural networks can easily learn required features. However, the accuracy of the overall optimized model reduces slightly in the case of cracks due to complex features for the crack class.

From the detection of crack detection based on size, we notice that large damages are easier to recognize compared to smaller cracks, which is due to more features present in larger crack. Further, we also notice that our algorithm is very robust to noises and has small false positives for no damages.

## 6. Conclusions

Conventional inspections of the continuous long conveyor belt have been performed visually. However, due to the high burden on the safety engineers and poor efficiency, there is a strong need to develop a system that automatically detects cracks and localize it without stopping the work. In this research, we develop a novel methodology to detect and localize conveyor belt cracks in real-time with offline server running [39] on edge devices. The CBCD dataset containing 9,362 images to detect both cracks and digits on the conveyor belt was developed to train a YOLOv4 model and optimize the original network by techniques such as layers fusion, precision reduction, etc. for carrying out inference on lightweight edge devices. The optimized model can detect cracks and digits with mAP of 0.850 and 0.99, respectively with 15 frames per second on edge device. Further, considering a fixed distance of the camera from the conveyor belt, size of the damage was estimated using a monocular camera to categorize the seriousness of the damage with respect to ongoing mucking work.

In the continuation of our research work, we would like to improve the crack detection for smaller crack sizes, which can be done by combining multiple frame detection [40] result into one due to good field of view from the camera. Research results presented in this paper could also be applied to other domains to mark the versatile nature of real time deep learning applications such as manufacturing and mining industries.

## Acknowledgments

We acknowledge Mr. Takeshi Hosokawa of Tokyo Kizai Kogyo Co., Ltd., and Mr. Tsuneo Koike of Ando Hazama Corporation for their cooperation and fruitful discussions.

## Author Contributions

**Data curation:** Uttam Kumar Dwivedi.

**Funding acquisition:** Uttam Kumar Dwivedi.

**Methodology:** Uttam Kumar Dwivedi.

**Project administration:** Uttam Kumar Dwivedi.

**Software:** Ashutosh Kumar.

**Supervision:** Yoshihide Sekimoto.

**Validation:** Uttam Kumar Dwivedi.

**Writing – original draft:** Uttam Kumar Dwivedi.

**Writing – review & editing:** Uttam Kumar Dwivedi.

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
