## [Decision Letter · Decision Letter 0]

14 Oct 2022

PONE-D-22-23772Realtime detection and categorization of longitudinal cracks in conveyor belt using deep-learning approachPLOS ONE

Dear Dr. Dwivedi,

Thank you for submitting your manuscript to PLOS ONE. After careful consideration, we feel that it has merit but does not fully meet PLOS ONE’s publication criteria as it currently stands. Therefore, we invite you to submit a revised version of the manuscript that addresses the points raised during the review process.

We look forward to receiving your revised manuscript.

Kind regards,

Brij Bhooshan Gupta

Academic Editor

PLOS ONE

Journal Requirements:

2. Please note that PLOS ONE has specific guidelines on code sharing for submissions in which author-generated code underpins the findings in the manuscript. In these cases, all author-generated code must be made available without restrictions upon publication of the work. Please review our guidelines at https://journals.plos.org/plosone/s/materials-and-software-sharing#loc-sharing-code and ensure that your code is shared in a way that follows best practice and facilitates reproducibility and reuse. New software must comply with the Open Source Definition.

"This study is supported by Tokyo Kizai Kogyo co. ltd. and University of Tokyo, Japan. The authors declare no conflict of interest. We acknowledge Mr. Takeshi Hosokawa of Tokyo Kizai Kogyo Co., Ltd. and Mr. Tsuneo Koike of Ando Hazama Corporation for their cooperation in filming the continuous belt conveyor of this research."

"This work was supported by the financial support from Tokyo Kizai Kogyo co. ltd. (http://www.tokyokizai.com/). The funders had no role in study design, data collection and analysis, decision to publish, or preparation of the manuscript."

7. Please amend the manuscript submission data (via Edit Submission) to include authors Ashutosh  Kumar and Yoshihide Sekimoto.   

Reviewers' comments:

Reviewer's Responses to Questions

**Comments to the Author**

1. Is the manuscript technically sound, and do the data support the conclusions?

Reviewer #1: Partly

Reviewer #2: Yes

2. Has the statistical analysis been performed appropriately and rigorously? 

Reviewer #1: No

Reviewer #2: Yes

3. Have the authors made all data underlying the findings in their manuscript fully available?

Reviewer #1: No

Reviewer #2: Yes

4. Is the manuscript presented in an intelligible fashion and written in standard English?

Reviewer #1: No

Reviewer #2: Yes

5. Review Comments to the Author

Reviewer #1: I found the paper a little difficult to read because of the inconsistent use of good language and the ambiguous way the topic was presented. Particularly in this work, the presentation quality needs to be improved. By fully rewriting the content, an accomplished English-language author may significantly improve this document. I have the following ideas to improve this essay:

-Describe your contribution better.

-The literature review is sufficient, but the authors should organise them more effectively.

-Review the following articles to strengthen the paper's technical foundation:

Handling Data Scarcity Through Data Augmentation in Training of Deep Neural Networks for 3D Data Processing, Improved Semantic Representation Learning by Multiple Clustering for Image-Based 3D Model Retrieval,Optimization of the Wake-Up Scheduling Using a Hybrid of Memetic and Tabu Search Algorithms for 3D-Wireless Sensor Networks, Unobtrusive academic emotion recognition based on facial expression using rgb-d camera using adaptive-network-based fuzzy inference system (ANFIS),Accelerating 3D medical volume segmentation using GPUs

– Carefully correct all errors in this paper.

-Improve the paper's connections and overall flow.

Reviewer #2: The Author has made an effort to proposed Realtime detection and categorization of longitudinal cracks in conveyor belt using deep-learning approach.

Title is needed to relook and make it more appropriate in the view of contribution. Further, in the abstract, author relook its flow and highlight the key contributions.

• Paper should be strictly according to the journal template.

• Improve the figure quality.

• Add some comparison results and discussion.

• Give a comparison from previous work.

• Check for grammar and spellings.

• Literature review section need to be added.

• Result and discussion part is required to be improved.

• State the novelty of your work.

• The overall manuscript should be checked for typos, syntax, and grammar to improve the quality of content flow and presentation

• More Number of References Should be added in your article

6. PLOS authors have the option to publish the peer review history of their article (what does this mean?). If published, this will include your full peer review and any attached files.

Reviewer #1: No

Reviewer #2: No

---

## [Author Response · Author response to Decision Letter 0]

5 Feb 2023

Dear Brij Bhooshan Gupta,

Academic Editor,

PLOS ONE.

Warm regards. I sincerely thank you for informing the referees’ reports on our initial article titled “Realtime detection and categorization of longitudinal cracks in conveyor belt using deep-learning approach” (PONE-D-22-23772). I thank you and both the referees for their valuable time and constructive remarks. Following their comments to the letters, we have significantly modified the article presenting a completely new method of conveyor belt damage identification. Please find below the answers to all the questions raised by the referees on a point-by-point format.

Reviewer: 1

Comment: I found the paper a little difficult to read because of the inconsistent use of good language and the ambiguous way the topic was presented. Particularly in this work, the presentation quality needs to be improved. By fully rewriting the content, an accomplished English-language author may significantly improve this document. I have the following ideas to improve this essay:

Reply: Thank you very much for your constructive criticism and suggestions. We have modified our manuscript in accordance with your suggestions. 

-Describe your contribution better.

Reply: The corrections have been incorporated and highlighted in the revised manuscript. 

-The literature review is sufficient, but the authors should organise them more effectively.

Reply: Following your suggestion, we have added a new literature review section in the article to make the manuscript more coherent. 

-Review the following articles to strengthen the paper's technical foundation:

Handling Data Scarcity Through Data Augmentation in Training of Deep Neural Networks for 3D Data Processing, Improved Semantic Representation Learning by Multiple Clustering for Image-Based 3D Model Retrieval, Optimization of the Wake-Up Scheduling Using a Hybrid of Memetic and Tabu Search Algorithms for 3D-Wireless Sensor Networks, Unobtrusive academic emotion recognition based on facial expression using rgb-d camera using adaptive-network-based fuzzy inference system (ANFIS),Accelerating 3D medical volume segmentation using GPUs

Reply: We have carefully reviewed the mentioned papers and cited them in our revised manuscript. 

– Carefully correct all errors in this paper.

Reply: We have carefully proofread our manuscript for submission. 

-Improve the paper's connections and overall flow.

Reply: We have revised our manuscript to present a clear outlook and subject flow. 

Reviewer: 2

Comment: The Author has made an effort to proposed Realtime detection and categorization of longitudinal cracks in conveyor belt using deep-learning approach.

Title is needed to relook and make it more appropriate in the view of contribution. Further, in the abstract, author relook its flow and highlight the key contributions.

• Paper should be strictly according to the journal template.

Reply: We have revised our manuscript as per the journal template. 

• Improve the figure quality.

Reply: We have improved figure quality. 

• Add some comparison results and discussion.

Reply: We have added comparison results and discussions. 

• Give a comparison from previous work.

Reply: We have added and highlighted in our main text. 

• Check for grammar and spellings.

Reply: We have carefully proofread for possible grammar and spelling errors. 

• Literature review section need to be added.

Reply: We have added and highlighted Literature review section. 

• Result and discussion part is required to be improved.

Reply: We have improved results and discussion sections to give better prospects to the reader. 

• State the novelty of your work.

Reply: We have highlighted the novelty in the revised text. 

• The overall manuscript should be checked for typos, syntax, and grammar to improve the quality of content flow and presentation

Reply: The typos, syntax and grammar have been improved. 

• More Number of References Should be added in your article

Reply: We have cited more references. 

 I once again sincerely thank you and both the referees for their time, effort and constructive suggestions. Addressing their concerns, we have thoroughly revised the manuscript and we hope it meets the high standard of PLOS One journal. Thank you very much.

Sincerely yours,

Uttam Kumar Dwivedi

---

## [Decision Letter · Decision Letter 1]

1 Mar 2023

PONE-D-22-23772R1Real-time classification of longitudinal conveyor belt cracks with deep-learning approachPLOS ONE

Dear Dr. Dwivedi,

Thank you for submitting your manuscript to PLOS ONE. After careful consideration, we feel that it has merit but does not fully meet PLOS ONE’s publication criteria as it currently stands. Therefore, we invite you to submit a revised version of the manuscript that addresses the points raised during the review process.

We look forward to receiving your revised manuscript.

Kind regards,

Brij Bhooshan Gupta

Academic Editor

PLOS ONE

Journal Requirements:

Reviewers' comments:

Reviewer's Responses to Questions

**Comments to the Author**

1. If the authors have adequately addressed your comments raised in a previous round of review and you feel that this manuscript is now acceptable for publication, you may indicate that here to bypass the “Comments to the Author” section, enter your conflict of interest statement in the “Confidential to Editor” section, and submit your "Accept" recommendation.

Reviewer #1: (No Response)

Reviewer #2: All comments have been addressed

2. Is the manuscript technically sound, and do the data support the conclusions?

Reviewer #1: Partly

Reviewer #2: Partly

3. Has the statistical analysis been performed appropriately and rigorously? 

Reviewer #1: No

Reviewer #2: Yes

4. Have the authors made all data underlying the findings in their manuscript fully available?

Reviewer #1: No

Reviewer #2: Yes

5. Is the manuscript presented in an intelligible fashion and written in standard English?

Reviewer #1: Yes

Reviewer #2: Yes

6. Review Comments to the Author

Reviewer #1: I found the topic of your research to be interesting and relevant. However, I have identified some issues that require your attention. Firstly, I recommend that you expand your literature review to include more recent and relevant sources. i suggest a few like: An edge-AI based forecasting approach for improving smart microgrid efficiency, A multimodal, multimedia point-of-care deep learning framework for COVID-19 diagnosis, Service orchestration of optimizing continuous features in industrial surveillance using big data based fog-enabled internet of things, A novel approach for phishing URLs detection using lexical based machine learning in a real-time environment

Also, ensure that your introduction clearly and effectively contextualizes your research question. Furthermore, I noticed some inconsistencies in the data presented, which need to be addressed. Please review your results and ensure that they are presented clearly.

Finally, I have observed some minor issues with grammar and syntax that need to be addressed.

Reviewer #2: The Author has incorporated all the suggestions given in first review with respect to the made an effort to proposed Realtime detection and categorization of longitudinal cracks in conveyor belt using deep-learning approach

7. PLOS authors have the option to publish the peer review history of their article (what does this mean?). If published, this will include your full peer review and any attached files.

Reviewer #1: No

Reviewer #2: No

---

## [Author Response · Author response to Decision Letter 1]

23 Mar 2023

Reviewer: 1

Comment: I found the topic of your research to be interesting and relevant. However, I have identified some issues that require your attention. Firstly, I recommend that you expand your literature review to include more recent and relevant sources. i suggest a few like: An edge-AI based forecasting approach for improving smart microgrid efficiency, A multimodal, multimedia point-of-care deep learning framework for COVID-19 diagnosis, Service orchestration of optimizing continuous features in industrial surveillance using big data based fog-enabled internet of things, A novel approach for phishing URLs detection using lexical based machine learning in a real-time environment

Also, ensure that your introduction clearly and effectively contextualizes your research question. Furthermore, I noticed some inconsistencies in the data presented, which need to be addressed. Please review your results and ensure that they are presented clearly.

Finally, I have observed some minor issues with grammar and syntax that need to be addressed:

Reply: Thank you very much for your constructive criticism and suggestions. We understand your concerns regarding the expansion of literature review with the mentioned articles. However the mentioned articles don’t fit in our current discussion and are out of context to our study of damage detection in conveyor belts or in construction sites. 

Throughout our introduction we focused on addressing the research question: “How can edge AI-based deep learning framework be used to detect and track damages on long conveyor belts in real-time without halting the mucking process and ensure safety and productivity at mountain tunnel construction sites?” We have clearly mentioned this in our revised manuscript and highlighted. 

 Also, we have followed and addressed your earlier comments, which mentioned the literature review being sufficient and in need of better organizations. So at this stage we request you to kindly consider the manuscript literature section as it is. This is in-sync with the Reviewer 2. 

 We have carefully analysed our method statement and results. We are quite confident on our presentation. However, if you would kindly be more precise on which part of data presentation contains inconsistencies, we are ready to address and answer. 

 We have carefully proofread the manuscript and checked it through the professional software as well for spellings and grammar. We are happy for further proofread formalities if mentioned specifically. 

Reviewer: 2

Comment: The Author has incorporated all the suggestions given in first review with respect to the made an effort to proposed Realtime detection and categorization of longitudinal cracks in conveyor belt using deep-learning approach

Reply: We are delighted to receive positive response. We are grateful for all your efforts. 

 I once again sincerely thank you and both the referees for their time, effort and constructive suggestions. Addressing their concerns, we have thoroughly revised the manuscript and we hope it meets the high standard of PLOS One journal. Thank you very much.

Sincerely yours,

Uttam Kumar Dwivedi

---

## [Decision Letter · Decision Letter 2]

10 Apr 2023

Real-time classification of longitudinal conveyor belt cracks with deep-learning approach

PONE-D-22-23772R2

Dear Dr. Dwivedi,

We’re pleased to inform you that your manuscript has been judged scientifically suitable for publication and will be formally accepted for publication once it meets all outstanding technical requirements.

Kind regards,

Brij Bhooshan Gupta

Academic Editor

PLOS ONE

Additional Editor Comments (optional):

Reviewers' comments:

Reviewer's Responses to Questions

**Comments to the Author**

1. If the authors have adequately addressed your comments raised in a previous round of review and you feel that this manuscript is now acceptable for publication, you may indicate that here to bypass the “Comments to the Author” section, enter your conflict of interest statement in the “Confidential to Editor” section, and submit your "Accept" recommendation.

Reviewer #1: All comments have been addressed

2. Is the manuscript technically sound, and do the data support the conclusions?

Reviewer #1: Yes

3. Has the statistical analysis been performed appropriately and rigorously? 

Reviewer #1: No

4. Have the authors made all data underlying the findings in their manuscript fully available?

Reviewer #1: Yes

5. Is the manuscript presented in an intelligible fashion and written in standard English?

Reviewer #1: Yes

6. Review Comments to the Author

Reviewer #1: The author has incorporated all the suggestions and properly answered all the query that have been raised due to review.

7. PLOS authors have the option to publish the peer review history of their article (what does this mean?). If published, this will include your full peer review and any attached files.

Reviewer #1: No

---

## [Editor Report · Acceptance letter]

11 Jul 2023

PONE-D-22-23772R2 

Real-time classification of longitudinal conveyor belt cracks with deep-learning approach 

Dear Dr. Dwivedi:

I'm pleased to inform you that your manuscript has been deemed suitable for publication in PLOS ONE. Congratulations! Your manuscript is now with our production department. 

Kind regards, 

on behalf of

Dr. Brij Bhooshan Gupta 

Academic Editor

PLOS ONE